# Absence of MHC-II expression by lymph node stromal cells results in autoimmunity

Juan Dubrot[1],*, Fernanda V Duraes[1],*, Guillaume Harlé[1],*, Anjalie Schlaeppi[1] , Dale Brighouse[1], Natacha Madelon[1,2], Christine Göpfert[3] , Nadine Stokar-Regenscheit[3], Hans Acha-Orbea[4], Walter Reith[1], Monique Gannagé[1,2], Stephanie Hugues[1] 

How lymph node stromal cells (LNSCs) shape peripheral T-cell responses remains unclear. We have previously demonstrated that murine LNSCs, lymphatic endothelial cells (LECs), blood endothelial cells (BECs), and fibroblastic reticular cells (FRCs) use the IFN-γ–inducible promoter IV (pIV) of the MHC class II (MHCII) transactivator CIITA to express MHCII. Here, we show that aging mice (>1 yr old) in which MHCII is abrogated in LNSCs by the selective deletion of pIV exhibit a significant T-cell dysregulation in LNs, including defective Treg and increased effector CD4[+] and CD8[+] T-cell frequencies, resulting in enhanced peripheral organ T-cell infiltration and autoantibody production. The proliferation of LN-Tregs interacting with LECs increases following MHCII up-regulation by LECs upon aging or after exposure to IFN-γ, this effect being abolished in mice in which LECs lack MHCII. Overall, our work underpins the importance of LNSCs, particularly LECs, in supporting Tregs and T-cell tolerance.

## Introduction

T-cell precursors undergo thymic negative selection, which ensures the elimination of developing T cells expressing TCR-recognizing self-Ags with excessive affinity. However, some autoreactive T cells escape this process of clonal deletion and exit the thymus to populate secondary lymphoid organs (SLOs). Therefore, additional mechanisms of T-cell tolerance are required in the periphery to avoid the development of autoimmunity. Among them, resting DCs, which constantly sample self-Ags in peripheral tissues and reach the draining LNs through the afferent lymph, present self-Ag–derived peptides to naive T cells. In the absence of danger, this phenomenon leads to clonal deletion, or anergy of autoreactive T cells (Steinman et al, 2003; Mueller, 2010). Alternatively, Tregs, by exhibiting suppressive immunoregulatory functions, can inhibit autoreactive T cells. Different subsets of Tregs have been described so far. Natural Tregs bear an autoreactive TCR, are induced in the thymus, and express the transcription factor Foxp3. Peripheral-induced Tregs can express Foxp3 or not, and differentiate in SLOs (Chen et al, 2003; Swee et al, 2009; Wirnsberger et al, 2011). Preservation of Treg function and biology is crucial for peripheral tolerance.

Lymph node stromal cells (LNSCs) have recently been promoted to the rank of new modulators of T-cell responses. After being considered for years as simple scaffolding, forming routes, and proper environment for Ag-lymphocyte encountering, we recently learned that they also impact both DC and T-cell functions. Lymphatic endothelial cells (LECs) promote DC entry into and T-cell egress from LNs (Sixt et al, 2005; Pham et al, 2010; Braun et al, 2011), whereas CCL19/CCL21–producing fibroblastic reticular cells (FRCs) control immune cells entry and proper localization into LNs (Link et al, 2007; Tomei et al, 2009). Blood endothelial cells (BECs) control T-cell homing to LNs (Bajenoff et al, 2003). In addition, LECs and FRCs are the major source of IL-7 in LNs, ensuring T-cell homeostasis. In inflammatory situations, however, LECs and FRCs produce nitric oxide to constrict T-cell expansion (Khan et al, 2011; Lukacs-Kornek et al, 2011; Siegert et al, 2011), whereas LECs further impair DC maturation in a contact-dependent fashion (Podgrabinska et al, 2009). In the context of peripheral tolerance, LNSCs, and in particular LECs and FRCs, ectopically express a large range of peripheral tissue Ags (PTAs), and further present PTA-derived peptides through MHC class I (MHCI) molecules to induce self-reactive CD8[+] T-cell deletion (Cohen et al, 2010; Fletcher et al, 2010, 2011; Tewalt et al, 2012). We have previously demonstrated that, in addition to inducing CD4[+] T-cell dysfunction by presenting peptide-MHC class II (MHCII) complexes acquired from DCs, LECs, BECs, and

[1]Department of Pathology and Immunology, School of Medicine, University of Geneva, Geneva, Switzerland   [2]Division of Rheumatology, Department of Internal Medicine, University Hospital Geneva, Geneva, Switzerland   [3]Institute of Animal Pathology, Department of Infectious Diseases and Pathobiology, Vetsuisse Faculty, University of Bern, Bern, Switzerland   [4]Department of Biochemistry, University of Lausanne, Epalinges, Switzerland

Correspondence: stephanie.hugues@unige.ch; Monique.Ghannage@unige.ch
*Juan Dubrot, Fernanda V Duraes, and Guillaume Harlé contributed equally to this work
Juan Dubrot's present address is Broad Institute of Massachusetts Institute of Technology and Harvard, Cambridge, MA, USA
Fernanda V Duraes's present address is Novartis Institutes for BioMedical Research, Basel, Switzerland
Anjalie Schlaeppi's present address is Max Planck Institute of Molecular Cell Biology and Genetics, Dresden, Germany

FRCs endogenously express MHCII molecules (Dubrot et al, 2014). Central tolerance of self-reactive CD4[+] T cells is partially mediated by thymic epithelial cells (TECs), in which MHCII molecules are loaded with peptides derived either from phagocytosis and processing of extracellular Ags (Stern et al, 2006), or from autophagy and endocytosis of intracellular Ags (Adamopoulou et al, 2013; Aichinger et al, 2013). Whether these pathways can be involved in MHCII-restricted Ag presentation by LNSCs, and impact peripheral self-reactive T-cell responses, is currently unknown.

Here, we have used genetically modified mice in which MHCII expression by non-hematopoietic cells is abrogated. Upon aging, and compared with their control counterparts, these mice exhibit an enhancement of spontaneous autoimmune processes, with enhanced T-cell activation in SLOs and effector T-cell infiltration in peripheral tissues, as well as the production of autoantibodies. In contrast, the Treg compartment is significantly impaired in SLOs. Furthermore, Rag2[−/−] mice transferred with T cell isolated from LN of aging MHCII-deficient LNSC mice displayed similar immunological and clinical perturbations compared with recipient injected with age-matched control T cells, suggesting a direct link between MHCII expressed by LNSCs and the appearance of T cell–mediated signs of autoimmunity. Accordingly, upon aging or IFN-γ treatment, LECs up-regulate MHCII molecules, and interact with Treg to promote their proliferation. This phenotype is abolished in mice deficient for MHCII expression in LECs. Altogether, we prove that MHCII expression by LNSCs have a manifest impact in peripheral tolerance. Notably, LECs support self-Ag–specific T cell peripheral tolerance by promoting Treg proliferation through MHCII-restricted Ag presentation.

# Results

### Expression of MHCII by LNSCs

Our previous work has demonstrated that LNSCs impact CD4[+] T-cell biology by presenting peptide-MHCII complexes acquired from DCs (Dubrot et al, 2014). Although these findings revealed a novel function of the lymphoid stroma as casual APC, whether endogenous MHCII Ag presentation by LNSCs can also impact T-cell responses remains to be determined. To abrogate cell-intrinsic pIV-mediated MHCII expression by LNSCs, we have used mice deficient for the IFN-γ–inducible pIV of CIITA (pIV[KO]) (Waldburger et al, 2001). As expected, steady-state MHCII expression by LECs, BECs, and FRCs was only partially abrogated in pIV[KO] mice compared with control mice (Fig 1A). Indeed, we recently described that MHCII expression by LNSCs partially results from a combination of both DC-acquired and CIITA promoter IV (pIV)–driven, endogenously expressed, MHCII molecules (Dubrot et al, 2014). DCs use the pI promoter of CIITA, and therefore, MHCII transfer to LNSCs is not abrogated following pIV deletion (Dubrot et al, 2014). However, pIV deletion led to the abrogation of endogenous MHCII expression by LECs, BECs, and FRCs. We confirmed this by analysing a second CTIIA-regulated gene, H2-M, the oligomorphic MHCII molecule involved in MHCII Ag loading. Ex vivo LECs, BECs, and FRCs express low H2-M and MHCII levels in the steady state, but substantially up-regulate

H2-M and I-A[b] molecules at both mRNA (Fig 1B) and protein (Fig 1C) levels after IFN-γ treatment (s.c. injection), suggesting a potential role for MHCII-restricted Ag presentation by LNSCs during inflammation. Both steady state and IFN-γ–inducible endogenous MHCII expressions were abrogated in pIV[KO] mice (Fig 1B and C).

### Abrogation of endogenous MHCII in LNSCs

MHCII expression in cortical TECs (cTECs) depends on pIV. Because of the absence of MHCII expression by cTECs, pIV[KO] mice consequently exhibit a defect in CD4[+] T-cell thymic positive selection, and lack peripheral CD4[+] T cells (Waldburger et al, 2003). Therefore, pIV[KO] mice were crossed with transgenic mice expressing CIITA under the keratin 14 (K14) promoter, which is active in cTECs (Laufer et al, 1996), but not in LNSCs (Fig 2A and B). MHCII expression by cTECs was efficiently restored in K14 CIITA tg × *Ciita pIV*[−/−] (referred as *K14tgpIV*[KO] mice) mice compared with pIV[KO] mice, and to a similar extent compared with cTECs isolated from WT or control (K14 Ciita tg × *Ciita pIV*[+/−], referred to as *K14tg*) mice (Fig 2C). As a consequence and as described before (Irla et al, 2008; Thelemann et al, 2014, 2016), *K14tgpIV*[KO] mice exhibit normal CD4[+] T-cell frequencies in SLOs (Fig 2D). We did not observe any significant expression of the K14 transgene in LNSCs purified from these mice (Fig 2B). However, to avoid any off-target effect of the K14 Ciita transgene in LNSCs or other cells, *K14tgpIV*[KO] mice were always compared with *K14tg* controls in all experiments. Importantly, MHCII expression by mTECs is not altered in pIV[KO] and *K14tgpIV*[KO] mice (Fig 2C), for the, respective, following reasons: first, mTECs express the pIII of CIITA in addition to pIV (Irla et al, 2008), and second, the K14 promoter is not active in terminally differentiated mTECs (Sukseree et al, 2012). Therefore, negative CD4[+] T-cell selection is unaffected in *K14tgpIV*[KO] mice. Accordingly, we have previously demonstrated that CD4[+] thymocytes and peripheral CD4[+] T cells that developed in *K14tgpIV*[KO] mice (6–10 wk-old) were indistinguishable from WT with respect to cell numbers and TCR V-β repertoire (Irla et al, 2008).

IFN-γ–induced endogenous and pIV-dependent MHCII up-regulation, which was efficiently abolished in LECs, BECs, and FRCs from *K14tgpIV*[KO] mice compared with control mice (Fig 2E). However, the expression of MHCII by other LN cells, i.e., different DC subtypes, B cells, or monocytes/macrophages, which rely on pI and/or pIII of CIITA (Waldburger et al, 2001), was found unaffected in *K14tgpIV*[KO] mice (Fig 2F). Altogether, our data demonstrate that endogenous MHCII expression is efficiently abrogated in LNSCs of *K14tgpIV*[KO] mice.

### Mice deficient for MHCII expression in LNSCs develop spontaneous signs of T cell–mediated autoimmunity

LNSCs have been so far described to function as tolerogenic Ag-presenting cells. Therefore, we tested whether the absence if MHCII expression by LNSCs would result in self-reactive T-cell tolerance breakdown and lead to immunopathology. Analysis of peripheral organs showed that the percentage of CD4[+] and CD8[+] T cells infiltrating the spinal cord (SC) and salivary glands (SGs) were markedly increased in *K14tgpIV*[KO] compared with *K14tg* control

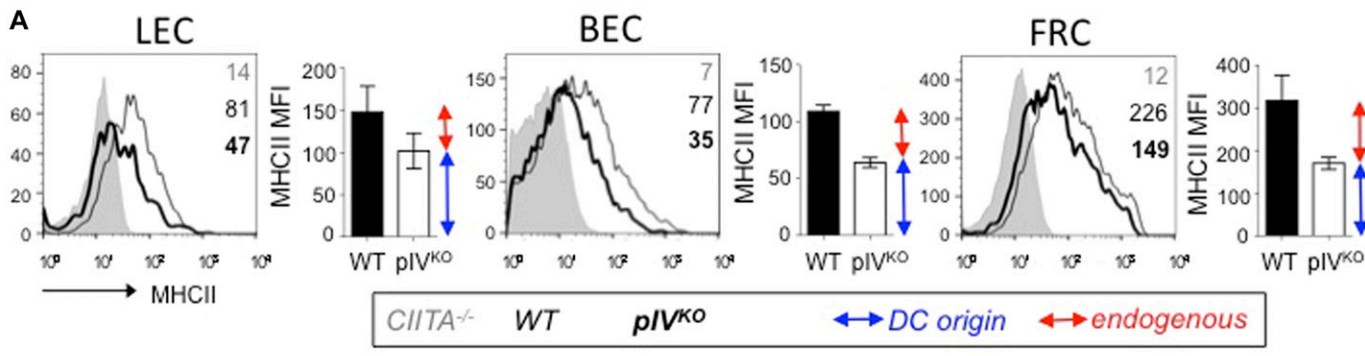

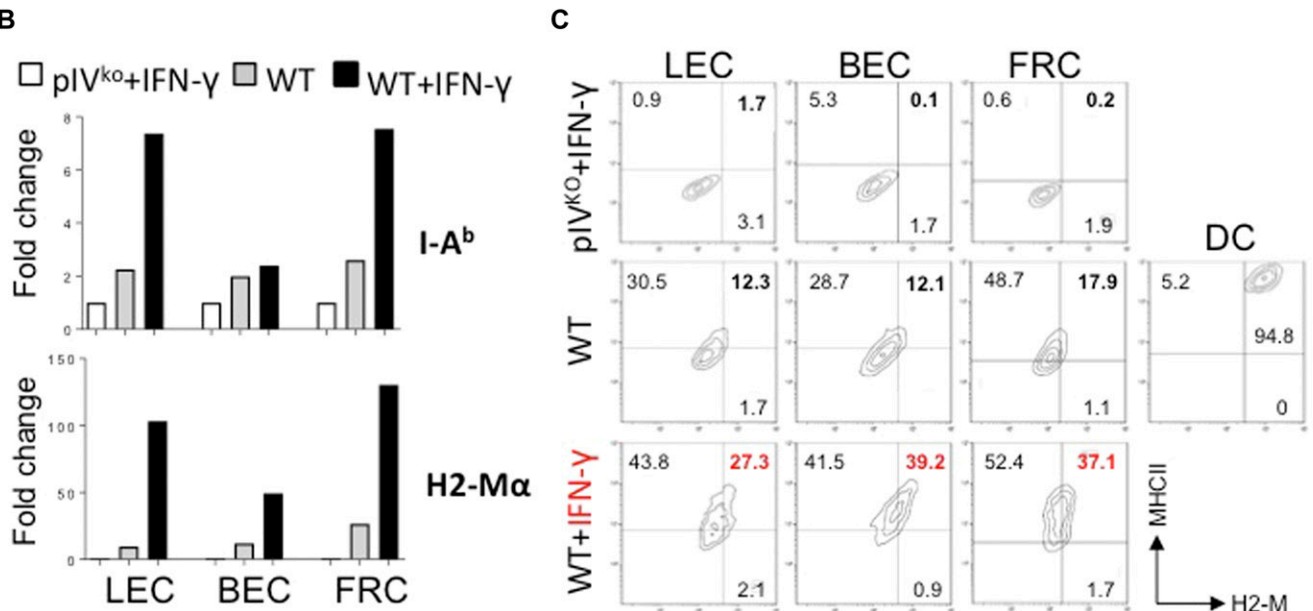

**Figure 1. LNSCs upte MHCII and MHCII-associated molecules in response to IFN-γ.**
**(A)** Flow cytometry and plotted histograms showing MHCII expression (MFI) by LECs, BECs, and FRCs from LN of indicated naive mice. MHCII coming from DC (blue arrow)- or endogenous (red arrow)-origin, is indicated. **(B)** WT or pIV CIITA–deficient mice (pIV$^{KO}$) mice were injected subcutaneously with PBS or 1 μg of IFN-γ; LNSCs were sorted by flow cytometry 24 h later. Histograms represent *I-A$^β$* (upper panel) and *H-2Mα* (lower panel) relative mRNA expression by LECs, BECs, and FRCs from indicated mice, measured by qPCR and normalized to GAPDH. Data are representative of two experiments with 10 mice pooled/group. **(C)** LNSCs and DCs from draining LNs were analysed 24 h after IFN-γ injection (as previously) by flow cytometry. Dot plots represent the expression of MHCII and H-2M by LECs, BECs, FRCs, and DCs from indicated mice. Frequency of double positive fractions are highlighted in red. Data are representative of two experiments with three mice/group.

elderly (18 mo-old) mice (Fig 3A). No difference was observed in 4-mo-old mice (Fig 3A). In addition, intestines of 18-mo-old *K14tgpIV$^{KO}$* exhibit increased frequencies of CD4$^+$ T cells producing the proinflammatory cytokine IFN-γ compared with *K14tg* controls (Fig 3B). Other peripheral organs, such as the liver and the lungs, also exhibit a tendency of enhanced infiltrating CD4$^+$ and CD8$^+$ T cell numbers in 18-mo-old *K14tgpIV$^{KO}$* mice, although not significant (not shown). Therefore, peripheral tissue T-cell infiltration was increased in a variety of non-lymphoid organs in elderly *K14tgpIV$^{KO}$* mice. In agreement with a possible autoimmune syndrome, serum obtained from 18-mo-old *K14tgpIV$^{KO}$* mice contained a broader spectrum of autoantibodies with enhanced reactivity to proteins from several tissues compared with *K14tg* control mice (Fig 3C), indicating an exacerbated development of systemic autoimmunity upon aging. Altogether, our data provide evidence that MHCII molecule expression by non-hematopoietic cells contribute

to peripheral T-cell tolerance by providing a brake in the development of spontaneous signs of T cell–mediated autoimmune inflammation in peripheral tissues of elderly mice.

### MHCII abrogation in LNSCs enhances T-cell activation and impairs the Treg compartment in LNs

We next sought to determine whether exacerbated signs of T cell–mediated autoimmune inflammation in peripheral tissues of elderly *K14tgpIV$^{KO}$* mice result from enhanced T-cell activation in SLOs. Analysis of CD4$^+$ and CD8$^+$ T cells in LN did not show any differences between 4-mo-old *K14tgpIV$^{KO}$* and *K14tg* mice (Fig 4A). However, in aging mice (18 mo), we noticed a significant enhanced T-cell activation, with decreased frequencies of naive (CD62L$^{hi}$CD44$^{lo}$) and increased frequencies of activated (CD62L$^{lo}$CD44$^{hi}$) CD4$^+$ and CD8$^+$ T cells in LNs of *K14tgpIV$^{KO}$* compared with *K14tg* mice (Fig 4A). In

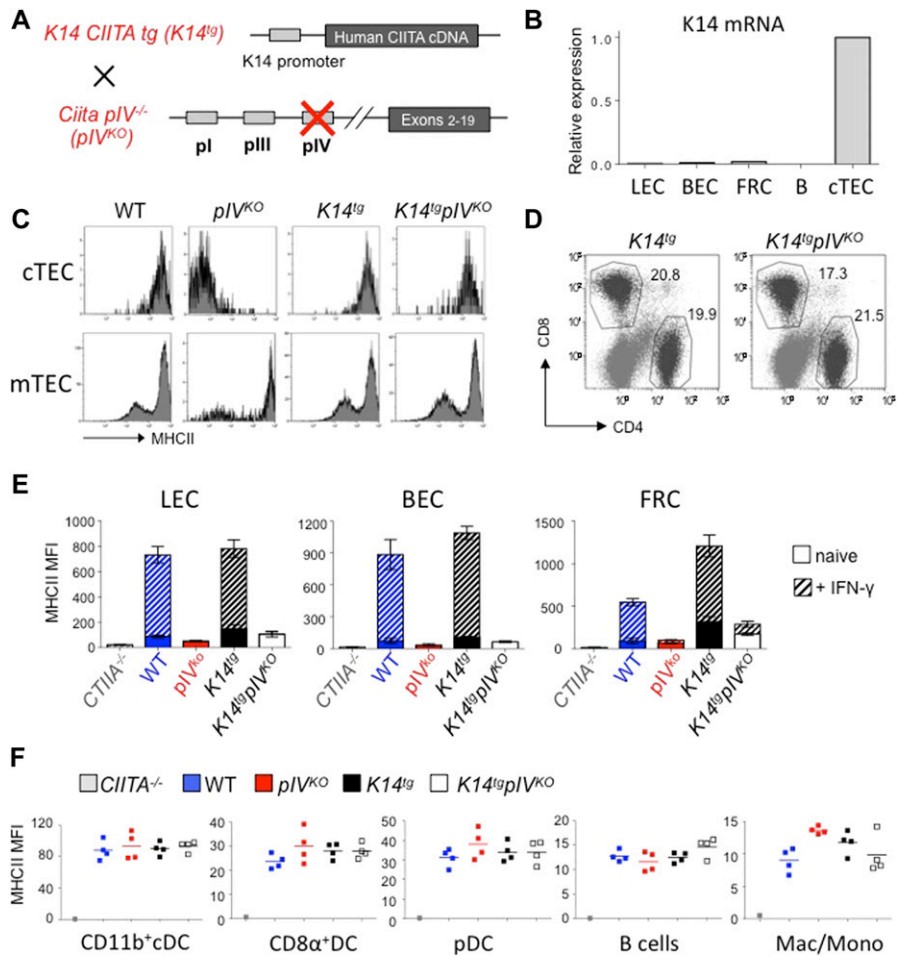

**Figure 2. Selective abrogation of endogenous MHCII expression in LNSCs.**
**(A)** Schematic representation of *K14tgpIV^KO* mice. Briefly, pIV^KO were crossed with transgenic mice expressing the full CIITA cDNA under the control of the keratin 14 (K14) promoter (*K14tg*). **(B)** Relative mRNA expression of K14 by LECs, BECs, FRCs, B cells and cTEC from *K14tgpIV^KO* mice, measured by qPCR and normalized to GAPDH. Data are representative of two experiments with 10 mice pooled/group **(C)** Flow cytometry histograms showing the expression of MHCII molecules by cTECs (gated on CD45^neg EpCAM^+Ly5.1^hi cells) and mTECs (gated on CD45^neg EpCAM^+Ly5.1^int cells) from indicated mice. **(D)** Flow cytometry dot plots showing CD8^+ and CD4^+ T-cell frequencies in LN of indicated mice (gated on CD3^+ cells). **(C, D)** Data are representative of two experiments with three mice/group. **(E)** Histograms showing MHCII expression (MFI) by LECs, BECs, and FRCs from LN of indicated mice, either naive (filled) or injected s.c. with IFN-γ 24 h before (hatched). **(F)** Graphs showing MHCII expression (MFI) by CD11b^+ cDCs (CD11c^hi CD11b^+), CD8α^+ DCs (CD11c^hi CD8α^+), pDCs (CD11c^int PDCA-1^+), B cells (CD19^+), and monocytes/macrophages (CD11c^neg CD11b^+) isolated from LNs of indicated mice. **(E, F)** Data are representative of two experiments with 3-4 mice/group.

addition, the frequency of effector CD4^+ and CD8^+ T cells expressing PD-1, and the frequency of CD4^+ and CD8^+ T cells producing IFN-γ, were enhanced in skin LNs from *K14tgpIV^KO* compared with *K14tg* mice (Fig 4B). Because both CD4^+ and CD8^+ T-cell phenotypes and effector functions were affected, we postulated that an active mechanism of T-cell inhibition might have been lost in LNs of *K14tgpIV^KO* mice upon aging. Accordingly, the frequency of Foxp3^+ Tregs was significantly impaired (two-fold reduction) in LNs of aging *K14tgpIV^KO* compared with *K14tg* control mice (Fig 4C). Foxp3^+ staining on LN sections confirmed this phenotype (Fig 4D). In contrast, no difference in Treg frequencies was observed in 4-mo-old mice (Fig 4C). With Treg frequencies being reduced in aging *K14tgpIV^KO* compared with *K14tg* controls, we next wondered whether this cellular population exhibited any phenotypic or functional alteration. We did not observe any alteration in the expression of several typical markers characterizing the Treg compartment, such as PD-1, CD44, CD25, or Foxp3 (not shown). In contrast, compared with control mice, LN Treg isolated from aging *K14tgpIV^KO* mice exhibited an impaired ability to suppress the proliferation of naive CD4^+ T cells in vitro at the Treg:Tnaive ratio of 1:5 (Fig 4E). The fact that no difference was observed when increased Treg numbers were used suggests that impaired Treg functions in knockout mice can be overcome with high Treg numbers.

So far, we have accumulated convincing data that upon aging, *K14tgpIV^KO* exhibits signs of spontaneous autoimmune defects. Organ-infiltrating T cells, and also the production of autoantibodies are increased, suggesting that the lack of MHCII expression by LNSCs in LNs of knockout elderly is likely to be responsible for this phenotype. However, these observations might also result from the lack of pIV of CIITA in the target organs, as other non-hematopoietic cells from peripheral tissues may up-regulate MHCII and contribute to this phenotype. Indeed, peripheral tissue-resident cells can up-regulate MHCII molecules in an IFN-γ–inducible pIV-dependent pathway and, by re-stimulating infiltrating T cells, possibly influence the outcome of T-cell responses (Duraes et al, 2013). Therefore, to determine whether enhanced signs of spontaneous autoimmunity in aging *K14tgpIV^KO* mice were a direct consequence of MHCII deficiency in LNSCs, we adoptively transferred T cells isolated from LN of *K14tgpIV^KO* or control *K14tg* mice into Rag2^−/− mice. 4 mo later, recipients that received T cells from >18-mo-old *K14tgpIV^KO* mice exhibited a significant loss of weight compared with mice injected with T cells isolated from age-matched *K14tg* controls (Fig 5A). Accordingly, immunofluorescence staining on small intestine sections revealed that Rag2^−/− hosts that received LN T cells from *K14tgpIV^KO* mice exhibited increased infiltrating CD45^+ cells, and in particular in the CD3^+ T-cell compartment (Fig 5B). In addition,

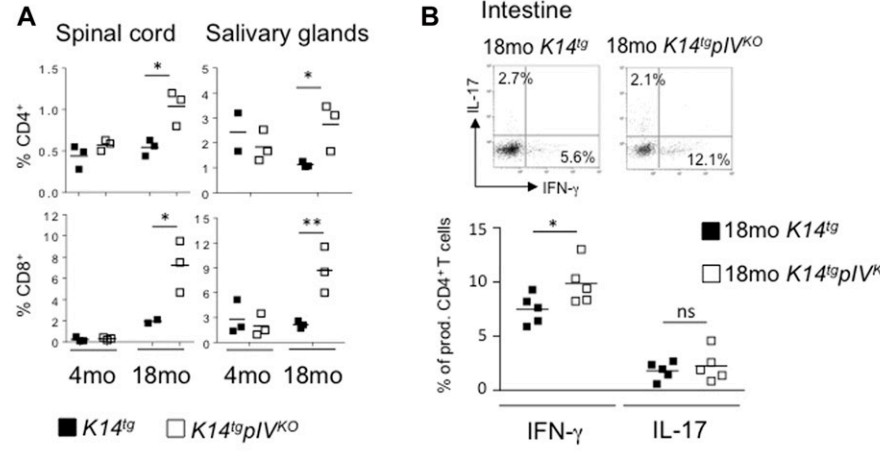

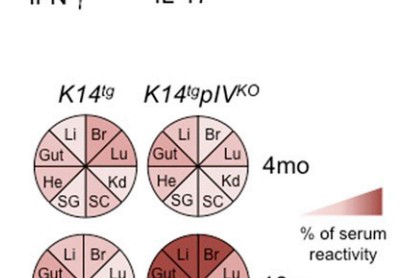

**Figure 3. Aging *K14tgpIV^KO* mice exhibit enhanced effector T-cell infiltration in peripheral organs and increased levels of autoantibodies.**
**(A)** *K14tgpIV^KO* (white) and *K14tg* control mice (black) were analysed at 4 and 18 mo. Percentages of infiltrating CD4⁺ (upper) and CD8⁺ (lower) T cells in SC and SGs from *K14tgpIV^KO* and control *K14tg* mice at the indicated ages. Data are pooled from five individual mice. **(B)** Flow cytometry analyses of cytokine production by infiltrating CD4⁺ T cells in the intestine of 18-mo-old *K14tgpIV^KO* and control *K14tg* mice. Data are pooled from four individual mice and are representative of two experiments. **(C)** Sera from 18-mo-old *K14tgpIV^KO* and control mice were individually tested for antibody reactivity against organ-specific proteins in immunoblots loaded from left to right; the liver (Li), brain (Br), lung (Lu), kidney (Kd), SC, SGs, heart (He), and Gut. Representative examples are shown. Colored circles represent the intensity of serum reactivity in the immunoblots. Data are pooled from five individual mice. Two-way Anova; *P < 0.05; **P < 0.01; n.s., not significant. Error bars depict mean ± SEM.

frequencies of naive CD62L^hiCD44^lo CD4⁺ and CD8⁺ T cells were decreased in LN of mice injected with T cells from *K14tgpIV^KO* donors (Fig 5C). Frequencies of effector CD4⁺ T cells producing IFN-γ or IL-17 were also augmented, as well as the frequency of IFN-γ producing CD8⁺ T cells, in LN of Rag2⁻/⁻ mice transferred with T cells from *K14tgpIV^KO* mice, (Fig 5D). Again, no significant difference was observed when adoptively transferred T cells were isolated from 4-mo-old *K14tgpIV^KO* or control *K14tg* donors (not shown). As observed in donor LNs, the frequency of Foxp3⁺ Tregs was significantly reduced in LNs of recipient Rag2⁻/⁻ mice injected with 18-mo-old *K14tgpIV^KO* compared with control *K14tg* T cells (Fig 5E), suggesting that the impaired Treg compartment would account for autoimmune T-cell disorders. Although a role for MHCII expression by cells in peripheral organs cannot be formally excluded, our data show that both Treg and non-Treg cells were affected in the absence of MHCII-restricted Ag presentation by LNSCs.

Altogether, our data suggest that MHCII expression by LNSCs contributes to self-reactive T-cell tolerance. When abrogated, it results in impaired Treg frequencies, enhanced self-reactive T-cell activation, and subsequent peripheral tissue infiltration in elderly mice.

### Deletion of MHCII on LNSCs decreases Treg proliferation

Interestingly, the levels of expression of MHCII molecules by LECs were significantly higher in elderly (18 mo) compared with young (4 mo) mice (Fig 6A). A slight, but not significant, increase in MHCII expression by BECs, but not by FRCs, was also observed. These observations can be a consequence of increased levels of IFN-γ in

old versus young LNs, (although undetectable by ELISA, not shown) and/or an enhanced sensitivity to IFN-γ. Accordingly, LECs and BECs express higher levels of IFN-γ receptor in LNs from elderly compared with younger mice (Fig 6A). These results suggest a more substantial contribution of MHCII-restricted Ag presentation by LECs in aging mice, and provide a possible explanation of why the phenotype observed in *K14tgpIV^KO* only appeared after several months.

Treg frequencies were found impaired in aging (18 mo), but not in young adult (4 mo) *K14tgpIV^KO* mice lacking MHCII expression in LNSCs. Therefore, we tested whether Treg proliferation may be affected by the loss of MHCII expression in LECs. Immunofluorescence staining on LN sections from *K14tgpIV^KO* and control elderly showed that the frequency of proliferating (red, Ki67⁺) Tregs (green, Foxp3⁺) in close interaction with LECs (Lyve-1⁺, in white) was significantly increased in aging controls compared with *K14tgpIV^KO* mice (Fig 6B).

We next tested whether IFN-γ stimulation, which promotes MHCII up-regulation in LNSCs, could accelerate this phenotype. WT mice were injected or not with IFN-γ, together with FTY720, which inhibits T-cell egress from LNs (Cyster & Schwab, 2012), and LECs and Treg were analysed in LNs 6 d after IFN-γ injection. Immunofluorescence staining revealed that the frequency of proliferating (red, Ki67⁺) Tregs (green, Foxp3⁺) in close interaction with LECs (Lyve-1⁺, in white) was significantly increased in mice treated with IFN-γ compared with untreated mice (Fig 6C). Again, to determine whether increased Treg expansion in LNs following IFN-γ injection was dependent on MHCII expression by LNSC, most likely by LECs, we performed similar experiments in *K14tgpIV^KO* and *K14tg* control

**Figure 4.** *K14tgpIV^KO* mice exhibit enhanced T-cell activation and impaired Treg frequencies in LNs upon aging.
*K14tgpIV^KO* (white) and *K14tg* control mice (black) were analysed at 4 and 18 mo. **(A)** Frequencies of naive (CD62L^hiCD44^lo) and activated/memory (CD62L^loCD44^hi) CD4^+ and CD8^+ T cells. **(B)** Frequencies of PD-1^+, IFN-γ, and IL-17 producing cells among CD4^+ and/or CD8^+ T cells. **(C, D)** Foxp3^+ Treg identification by flow cytometry (CD4^+ CD25^+ Foxp3^+) (C) and IHC staining (Foxp3^+) (D) in lymph nodes of indicated mice. **(A–D)** Data are representative of three experiments with 3–7 mice/group. *P < 0.05; **P < 0.01; ***P < 0.001; n.s., not significant. Error bars depict mean ± SEM. **(E)** CD4^+ CD25^hi T cells (Treg) from total skin LNs of 18-mo-old *K14tgpIV^KO* and *K14tg* mice were cultured (ratio 1:5, 1:3, and 1.1) with proliferation dye-labeled naive CD4^+ CD25^neg T cells. Labeled T-cell proliferation was assessed by flow cytometry after 5 d of co-culture. Data are representative of two experiments. Two-way Anova; **P < 0.01.

mice. We observed an increased Treg proliferation in LNs of *K14tg* controls, and further demonstrated that proliferating (red, Ki67^+) Tregs (green, Foxp3^+) (red arrows) in close proximity of LECs (white, Lyve-1^+) (Fig 6C). In contrast, Treg proliferation in LEC proximity (red arrows) was lower in LNs of *K14tgpIV^KO* mice, demonstrating that the increase of proliferation of Tregs interacting with LECs was dependent on their expression of MHCII (Fig 6C). In contrast, we did not observe any increase in the proliferation of non-Treg (Foxp3^neg) CD4^+ T cells upon IFN-γ stimulation, nor difference between *K14tg* controls and *K14tgpIV^KO* mice (not shown), suggesting that the Treg compartment is specifically affected. Finally, to examine a potential direct contribution of LECs as MHCII-restricted APCs in inducing Treg proliferation, we repeated the above experiments in mice in which MHCII expression was selectively abrogated in LECs. For that, we used Prox-1-Cre^ERT2 mice, expressing the Tamoxifen-inducible Cre recombinase under the promoter Prox-1, which is selectively expressed in adult LECs (Bazigou et al, 2011). Prox-1-Cre^ERT2 mice were crossed with MHCII^fl mice, allowing the selective deletion of MHCII molecules in LECs upon Tamoxifen treatment (not shown). Immunofluorescence analyses of LN sections demonstrated that

the frequency of proliferating Tregs in contact with LECs 6 d after IFN-γ injection was significantly reduced in mice in which LECs do not express MHCII molecules (Prox-1-Cre^ERT2 MHCII^fl) compared with control mice (MHCII^fl) (Fig 6D). Altogether, our results demonstrate that IFN-γ–mediated MHCII up-regulation by LNSCs promotes Treg proliferation, with a major contribution of LECs.

## Discussion

Our results provide the first evidence for an in vivo role of LNSCs in impacting peripheral T-cell tolerance as MHCII-restricted APCs. We have recently published that subsets of murine LNSCs present MHCII–peptide complexes acquired from DCs to induce CD4^+ T-cell dysfunction (Dubrot et al, 2014). Here, we further show that LNSCs inhibit autoreactive T-cell responses by directly presenting Ags through endogenous MHCII molecules. A recent study suggested that, although LECs express surface MHCII molecules, they cannot present a model Ag to CD4^+ T cells presumably because of the lack of expression of H2-M in steady state, preventing the

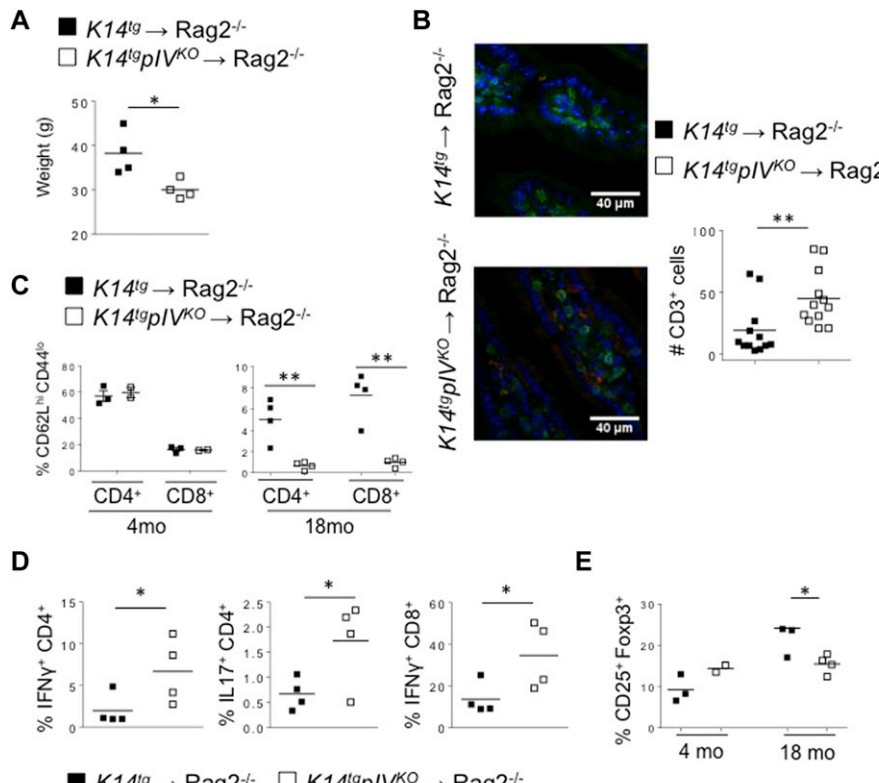

**Figure 5. T cells isolated from mice lacking MHCII expression in LNSCs induce autoimmunity upon transfer into immunodeficient mice.**
(A–E) T cells isolated from LNs and spleen of 18-mo-old *K14tgpIV*[KO] (white) and control mice (black) were adoptively transferred into Rag2[−/−] recipient mice that were analysed 4 mo later. (A) Mouse weight. Data are pooled from four mice and representative of two experiments. (B) Sections of small intestines from recipient Rag2[−/−] mice were stained with antibodies against CD45 (green), CD3 (red), and with DAPI (blue). Pictures show representative tissue sections. Graphs represent the quantification of CD3[+] T cells/area and are pooled from 12 tissue sections from three individual mice/group. (C–E) Flow cytometry dot plots showing the frequency of CD62L[hi]CD44[lo] CD4[+] and CD8[+] T cells (C), the frequency of CD4[+] T cell–producing IFN-γ or IL-17 and the frequency of CD8[+] T cell–producing IFN-γ in LNs (D) and Treg frequencies (E) in LNs of transferred Rag2[−/−] recipient mice. Data are representative of two experiments with 2–4 mice/group each. (C–E) *P < 0.05; **P < 0.01; ***P < 0.001; n.s. (A, B, D) unpaired *t* test, (C, E) two-way Anova.

loading of endogenous antigenic peptides onto MHCII molecules (Rouhani et al, 2015). However, we show that LECs, BECs, and FRCs that express the IFN-γ–inducible-CIITA pIV, require IFN-γ to up-regulate H-2M molecules, as they do for MHCII expression, these two genes being co-regulated by CIITA (Reith & Mach, 2001). In addition, LNSCs were described to express considerable amounts of invariant chain (Ii) and cathepsin L (Rouhani et al, 2015), and therefore seem well equipped for the presentation of antigenic peptides through MHCII in situations involving the presence of IFN-γ.

Genetic abrogation of MHCII in LNSCs in vivo leads to impaired regulatory T-cell frequencies and in vitro functions, enhanced effector T cell differentiation LNs and peripheral tissue infiltration, and subsequent development of T cell–mediated autoimmunity in elderly mice. Indeed, mice lacking MHCII in LNSCs exhibit spontaneous signs of autoimmunity in elderly mice. LNSCs express a broad range of self-Ags. Therefore, our hypothesis is that abrogation of MHCII in these cells will lead to an inflation of polyclonal memory/effector T cells. In agreement, no difference was observed in the frequency of several TCR Vβ chains expressed by CD4[+] and CD8[+] T cells isolated from LNs of old control and knockout mice (not shown), suggesting that there is no restriction of T-cell clonality in mice lacking MHCII in LNSCs. Importantly, in this model there is no pre-existing early CD4[+] T-cell bias, nor Treg function impairment. K14tgpIV[KO] young adult mice do not present any abnormality, excluding the possibility of a defect from thymic selection. In addition, T cells from 6- to 10-wk-old *K14tgpIV*[KO] mice were recently extensively characterized, and normal distribution of CD4[+] naive,

effector, and Foxp3[+] T-cell subsets were observed in the thymus, spleen, and peripheral LNs (Thelemann et al, 2014).

In contrast to a recent study suggesting an activity of K14 promoter in LNSCs (Baptista et al, 2014), we did not observe any K14 mRNA expression in LECs, BECs, and FRCs (Fig 2B). Consistently, MHCII expression was similar in LNSCs isolated from pIV[KO] and *K14tgpIV*[KO] mice (Fig 2E), confirming that the K14 ciita transgene does not promote any MHCII expression in LNSCs. However, because peripheral non-hematopoietic tissue-resident cells can up-regulate MHCII molecules in an IFN-γ–inducible pIV-dependent manner (Duraes et al, 2013), this mouse model does not represent per se a valid mouse model to study the specific role of endogenous MHCII expression by LNSCs in shaping peripheral CD4[+] T-cell responses. For instance, intestinal epithelial cells (IECs), hepatocytes and liver sinusoidal endothelial cells, pancreatic β cells, or astrocytes were postulated to be capable of MHCII-mediated Ag presentation, although very little in vivo data are available on the potential impact on peripheral CD4[+] T-cell responses (reviewed in Duraes et al [2013]). Recently, using the mouse model we are presently exploiting, two studies have highlighted opposite roles for MHCII expression in distinct target tissues. First, IFN-γ–mediated MHCII up-regulation by IECs exerts anti-inflammatory effects and protect against colitis (Thelemann et al, 2014). In contrast, in the context of experimental autoimmune myocarditis, elevated IFN-γ–inducible cardiac endothelial MHCII expression exacerbates the disease (Thelemann et al, 2016). Together, these findings provide evidence of a critical role of pIV-mediated non-hematopoietic MHCII expression in peripheral tissues

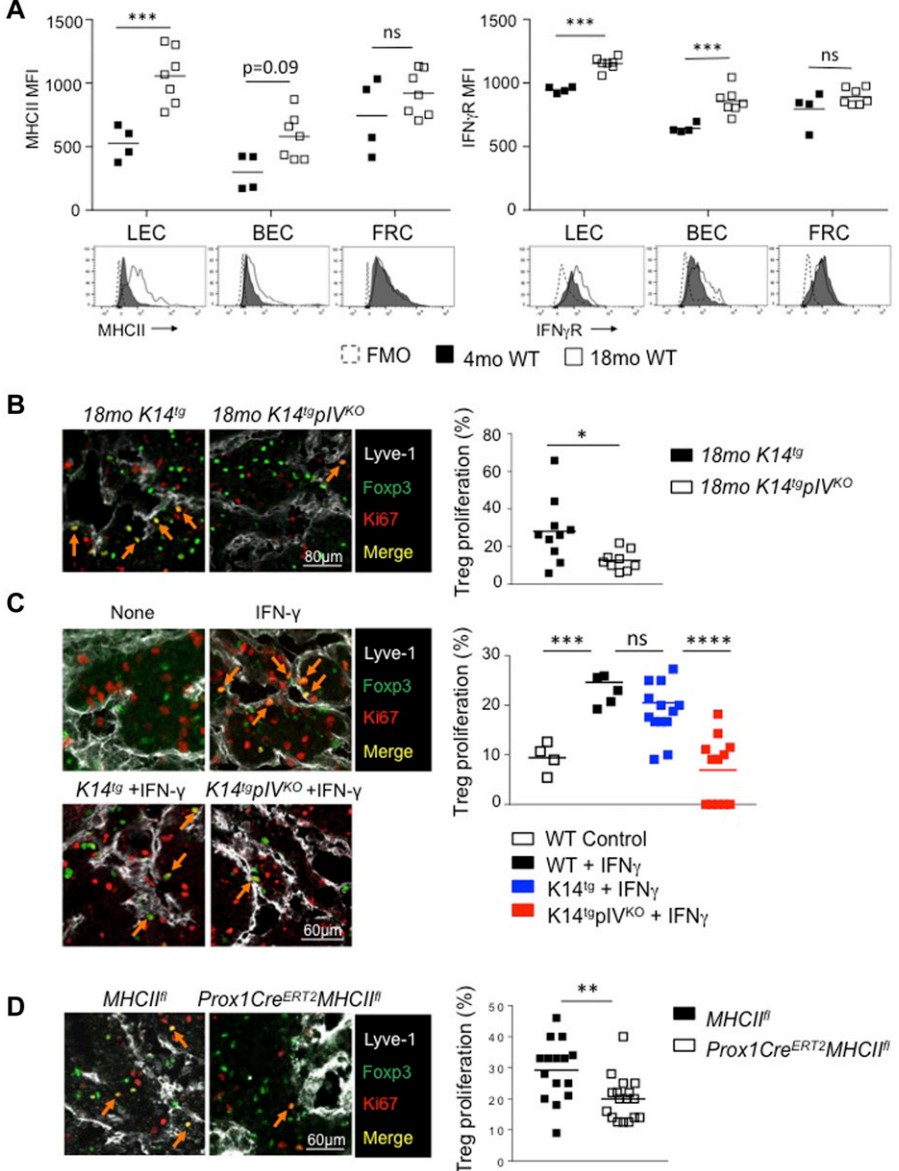

**Figure 6. Deletion of MHCII on LECs decreases Treg proliferation.**
**(A)** MHCII and IFN-γ receptor (IFNγR) expression (MFI) by LECs, BECs, and FRCs from LN of WT mice of the indicated age. Data are representative of two experiments with three mice/group each. **(B–D)** LN sections depicting LECs (Lyve-1, white) and proliferating Tregs (red arrows) (Ki67, red; and Foxp3, green) from 18-mo-old K14$^{tg}$ and K14$^{tg}$pIV$^{KO}$ mice (B), 3-mo-old WT and K14$^{tg}$ and K14$^{tg}$pIV$^{KO}$ (C), and 3-mo-old Tamoxifen-treated Prox-1-Cre$^{ERT2}$ MHCII$^{fl}$ and MHCII$^{fl}$ control mice (D). In (C) and (D), animals were treated with FTY720 every day for the last 6 d, and with IFN-γ 6 d before harvesting the LNs. Graphs represent the percentages of Ki67$^+$ among Foxp3$^+$ cells in contact with LECs from LN of the indicated mice. **(B–D)** Data are pooled from at least nine tissue sections from three individual mice/group. *P < 0.05; **P < 0.01; ***P < 0.001; n.s. (A, C) Two-way Anova, (B, D) unpaired *t* test.

upon inflammation, the impact on T-cell responses being largely dependent on the target organ. Here, we add a distinct role of LN-resident stromal populations in shaping CD4$^+$ T-cell responses, providing support to Treg homeostasis and maintaining peripheral tolerance. We cannot totally exclude a role for a lack of MHCII expression by cells in peripheral tissue on local T-cell reactivation in *K14tgpIV$^{KO}$* mice. Indeed, it is possible that T cells infiltrating peripheral organs are maintained in a tolerogenic state by local MHCII$^+$ cells, and that this phenomenon is abrogated in our mouse model. However, it is unlikely, for several reasons. First, a similar phenotype was obtained after adoptive transfer of T cells from LN of K14tgpIV$^{KO}$ mice into Rag2$^{-/-}$ with competent MHCII peripheral tissues. Importantly, T-cell compartments in the recipient Rag2$^{-/-}$ mice recapitulated the effects observed in the donor transgenic mice indicating a contribution of LNSCs to peripheral T-cell alterations. Second, although a contribution of MHCII expression by

tissue-resident cells has been described before, it was always in inflammatory settings, in which these cells indeed up-regulate MHCII in contrast to our stud which was performed in the steady state (Duraes et al, 2013), minimizing their possible contribution as MHCII-restricted Ag presenting cells in the absence of model-induced inflammation. Finally, LN-resident Tregs exhibit impaired proliferation in LNs not only in *K14tgpIV$^{KO}$*, but also in Prox-1-Cre$^{ERT2}$ MHCII$^{fl}$ mice, supporting a pro-tolerogenic role for MHCII-restricted Ag presentation by LNSCs, in particular LECs, in dampening T-cell autoimmune reactions.

We have previously shown that endogenous MHCII expression by LECs, BECs, and FRCs was significantly increased in WT mice compared to IFN-γ receptor–deficient mice (Dubrot et al, 2014), demonstrating that the presence of low levels of IFN-γ in naive mice drives pIV-mediated MHCII endogenous expression in steady-state LNSCs. This was confirmed by reduced MHCII expression by LNSCs in steady-state

LNs from *K14tgpIV^KO* compared with control mice. Therefore, basal (and presumably variable) levels of circulating steady-state IFN-γ promote endogenous MHCII expression by LNSCs, and confer them the ability to function as competent MHCII-mediated APCs. Increased inflammatory status over time might be due to the accumulation of punctual and mild infectious processes or altered gut microbiotic composition upon aging (Claesson et al, 2012).

Last, we provide evidence for a direct effect of MHCII-restricted Ag presentation by LECs in promoting Treg proliferation. Genetic selective abrogation of MHCII expression by LECs in mice results, similarly to what was observed in K14tgpIV^KO mice, in impaired proliferation of Treg in contact with LECs. Therefore, among the LNSC subsets, LECs seems to be the main contributors in inducing Treg proliferation. Distinct sets of PTA expression, together with differential levels of expression of co-inhibitory molecules, such as PDL-1 (Tewalt et al, 2012), by LNSC populations might give rise to distinct impacts on T-cell responses, such as T-cell apoptosis, T-cell anergy, or Treg induction. Elevated PDL-1 expression by LECs compared with BECs and FRCs might explain a role for those cells in impacting Tregs, which express high levels of PD-1. Although we did not elucidate the molecular mechanisms, a selective effect on Tregs, which mostly express a self-reactive TCR, could result from the ability of LECs to present endogenously expressed PTAs. One explanation could be the use of the autophagy pathway by LECs to present endogenous PTAs through MHCII molecules. This hypothesis would be in accordance with previous results obtained in the thymic epithelium (Nedjic et al, 2008; Aichinger et al, 2013). Future experiments in the laboratory will use mice genetically deficient for autophagy to firmly demonstrate the implication of this pathway in the ability of the different LNSC subsets to present PTAs through MHCII molecules and to alter the Treg compartment. Our experiments do not rule out another possible mechanism of action by which MHCII expression in LNSC may support T-cell inactivation in an Ag-independent fashion. MHCII is a natural ligand for the molecule lymphocyte activation gene 3 (LAG-3), expressed on activated T cells and can be induced by IFN-γ stimulation. This immune checkpoint has an inhibitory role for LAG-3 in controlling both CD4 and CD8 T-cell proliferation in vitro and in vivo (Workman et al, 2004; Grosso et al, 2007). In addition, LAG-3 expression on Tregs has been shown to be important for their suppressive function (Grosso et al, 2007; Okamura et al, 2009). Moreover, the co-expression of LAG-3 with the inhibitory receptor PD-1 on exhausted T cells or Tregs correlates with a greater state of effector T-cell exhaustion and the suppressive function of Tregs. The relative contribution of both Ag-dependent and/or Ag-independent pathways remains to be addressed.

In conclusion, our work identifies novel in vivo functions of MHCII expression by LNSCs in the maintenance of peripheral T-cell tolerance by inhibiting autoreactive T cells. We also define the LECs as important MHCII expressing cells supporting Treg homeostasis.

# Materials and Methods

## Mice and treatments

C57BL/6 WT mice were purchased from Charles River. pIV^−/− (Waldburger et al, 2003), Ciita^−/− and *K14tgpIV^KO* (Laufer et al, 1996),

and Rag2^−/− (Jackson Laboratories) Prox-1-Cre^ERT2 MHCII^fl mice have been previously described (Bazigou et al, 2011). To selectively abrogate MHCII expression in LECs, Prox-1-Cre^ERT2 MHCII^fl/fl (and MHCII^fl/fl control) mice were injected with tamoxifen (i.p) twice a day for four consecutive days (2 mg/mice/d). 2 wk after the last injection, mice were treated with IFN-γ and FTY720 as described below. These mouse strains are on a C57BL/6 background and were housed and maintained under SPF conditions at the Geneva Medical School animal facility and under EOPS conditions at Charles River, France. All animal husbandry and experiments were approved by and performed in accordance with guidelines from the Animal Research Committee of the University of Geneva. Genotyping was carried out by using PCR on ear biopsies with the Phire Tissue Direct PCR Master Mix (F170L; Thermo Fisher Scientific). When indicated, IFN-γ (Peprotech) was injected subcutaneously (both flanks and neck, 1 μg/50 μl), and FTY720 (Sigma-Aldrich) (20 μg) was injected every day for 6 d before the experiment. Mice from each group were randomized before treatments.

## LNSC and cell tissue isolation

LNSCs were obtained as previously described (Dubrot et al, 2014). In brief, total skin or skin-draining LNs from individual or 10–15 pooled mice were cut into small pieces and digested in RPMI containing 1 mg/ml collagenase IV (Worthington Biochemical Corporation), 40 μg/ml DNase I (Roche), and 2% FBS. Undigested cells were further digested with 1 mg/ml collagenase d, and 40 μg/ml DNase I (Roche). The reaction was stopped by addition of 5 mM EDTA and 10% BSA. Samples were further disaggregated through a 70-μm cell strainer and blocked with anti-CD16/32 antibody. Single-cell suspensions were negatively selected using CD45 microbeads and a magnetic bead column separation (Miltenyi Biotec).

Cells from LN, the spleen, the SC, and SGs were isolated by digesting organ fragments with an enzymatic mix containing collagenase D (1 μg/ml) and DNAse I (10 μg/ml) (Roche) in HBSS [14]. For SC and SG, single-cell suspensions were further centrifuged through a discontinuous 30:70% percoll (Invitrogen) gradient. T cells were further analysed by flow cytometry.

Small intestines were excised and transferred into PBS 2% FCS. Peyer's patches and adipose tissue were carefully removed. The small intestines were cut longitudinally and cleaned of faecal content. The small intestines were next incubated in RPMI containing 10% FCS, 2 mM EDTA, and 1 mM DTT for 30 min at 37°C in a shaking incubator to remove epithelial cells, the supernatant being discarded. The remaining tissue was digested twice in RPMI containing 1 mg/ml of collagenase D (Roche) for 30 min at 37°C in a shaking incubator. Cells were recovered from the supernatant and filtered through a 70-μm cell strainer.

## Antibodies, flow cytometry, and cell sorting

Anti-gp38 (8.1.1), anti-CD31 (390), anti-CD11c (N418), anti-CD44 (IM7), and anti-IAb (AF6.120.1) mAbs were from BioLegend. Anti-CD45 (30F11), anti-CD16/32 FcγRIII (2.4G2), anti-I-A^d/I-E^d (2G9), anti-IFNγR (GR20), and H2-M (2E5A) were from BD. Anti-CD19 (1D3), anti-CD8 (53–6.7), anti-CD4 (GK1.5), anti-CD11b (M1/70), anti-CD62L (MEL-14), anti-PDCA-1 (eBio927), anti-PD-1 (J43), anti-IFN-γ (XMG1.2),

anti-IL-17 (eBio17B7), and anti-Foxp3 (FJK-16s) were from eBio-science. Anti-EpCAM (caa7-9G8) and anti-Ly5.1 (6C3) were from Miltenyi Biotec.

For LNSC flow cytometry sorting, enriched CD45$^{neg}$ cells were stained with mAbs against CD45, gp38, and CD31. In some experiments, cells were also stained with mAbs against MHCII or isotype control as indicated. For cTEC and mTEC sorting, enriched CD45$^{neg}$ cells were stained with mAbs against CD45, EpCAM, and Ly5.1.

Intracellular cytokine stainings were carried out with the Cytofix/Cytoperm kit (BD) for IFN-γ and IL-17 staining. Foxp3 staining was performed with the eBioscience kit, according to manufacturer's instructions. For IFN-γ and IL-17 staining, LN cells were cultured in RPMI containing 10% heat-inactivated fetal bovine serum, 50 mM 2-mercaptoethanol, 100 mM sodium pyruvate, and 100 $\mu$M penicillin/streptomycin at 37°C and 5% $CO_2$. Cells were stimulated for 18 h with PMA/ionomycin and Golgi stop solution (BD) was added during the last 4 h of culture before the staining.

Cells were either acquired on a Gallios or sorted using a MoFlowAstrios (Beckman Coulter), and analysed using FlowJo (Tree Star) or Kaluza softwares.

### Immunofluorescence microscopy

Mice were transcardiacally perfused with PBS, and intestines were fixed in paraformaldehyde before inclusion in paraffin. Preserved organs were cut into 7-$\mu$m-thick sections and deparaffinised in xylene–ethanol. Ag retrieval was performed in citrate buffer. Sections were then stained using labelled antibodies against CD45 (30-F11) and CD3 (17A2) and DAPI (Sigma-Aldrich) counter-staining. Sections were mounted with Mowiol fluorescent mounting medium (EMD). Images were acquired with a confocal microscope (LSM 700; Carl Zeiss Inc. and SP5; Leica).

Skin LNs from untreated mice or mice treated with IFN-γ and FTY720 were frozen in OCT medium. 10-$\mu$m-thick sections were cut and fixed with paraformaldehyde 4% for 20 min. After washing and permeabilization, the sections were stained overnight at 4°C using a rabbit anti-Lyve-1 antibody (Reliatech GmbH). Secondary staining was performed using a Alexafluor546-labelled donkey anti-rabbit antibody, Alexafluor488-labelled anti-Foxp3 (150D) antibodies and eF660-Ki67 (SolA15), for 2 h at room temperature. After DAPI (Sigma-Aldrich) staining, sections were mounted with Mowiol fluorescent mounting medium (EMD). Images were acquired with a confocal microscope (LSM 700; Carl Zeiss Inc. and SP5; Leica). Tregs in "close proximity" (<5 $\mu$m) of LECs were quantified as proliferating (Ki67$^+$Foxp3$^+$) or non-proliferating (Foxp3$^+$) cells, and the ratio of proliferating Tregs was calculated. At least 20 images/condition were quantified.

### Treg suppression assay

In vitro Treg suppressive assays were performed as follows: CD4$^+$ CD25$^{hi}$ T cells (Treg) were purified by flow cytometry from total skin LNs of 18-mo-old K14tgpIV$^{KO}$ and K14tg control mice and cultured at the indicated ratio with 2 × 10$^5$ proliferation dye-labeled naive CD4$^+$ CD25$^{neg}$ T cells in the presence of bone marrow–derived DC and anti-CD3 antibodies. Labeled T-cell proliferation was assessed by flow cytometry after 5 d of co-culture.

### T-cell adoptive transfer into Rag2$^{-/-}$ recipients

LN and spleen of donor mice were harvested, and T cells were negatively selected from the cell suspension using a PAN T isolation kit (Mylteniy biotec) according to the manufacturer's instructions. Purity generally exceeded 95%. 2–5 × 10$^6$ T cells were injected intravenously into Rag2$^{-/-}$ mice. Recipient mice were randomized before transfer and co-housed during the experiment.

### RNA isolation and quantitative RT-PCR

Total RNA was isolated using Tri-Reagent (Ambien) from sorted cells. cDNA was synthesized using random hexamers and M-MLV reverse transcriptase (Promega). PCRs were performed with the CFX Connect real-time PCR detection system and iQ SYBR green super mix (Bio-Rad Laboratories). The results were normalized with GAPDH or 60S ribosomal protein L32 mRNA expression and quantified with a standard curve generated with serial dilutions of a reference cDNA preparation. Primer sequences: I-Ab forward, 5'-CTG TGG TGG TGG TGA TGG T-3' and reverse, 5'-CGT TGG TGA AGT AGC ACT CG-3'. H2-Mα forward, 5'-CTCGAAGCATCTACACCAGTG-3' and reverse, 5'-TCCGAGAGCCCTATGTTGGG-3'. K14 orward, 5'-AGG GAG AGG ACG CCC ACC TT-3' and reverse, 5'- CCT TGG TGC GGA TCT GGC GG-3'

### Lysate preparation and immunoblotting

Tissues were homogenized in T-PER tissue protein extraction reagent (Thermo Fisher Scientific) supplemented with a cocktail of protease inhibitors (Complete, Roche). Protein extracts were incubated at 95°C for 5 min with of 4× SDS–PAGE–loading buffer (250 mmol Tris–HCl at pH = 6.8, 40% glycerol, 8% SDS, 0.57 mol β-mercaptoethanol, and 0.12% bromophenol blue). Equal amounts of protein were run on 12% SDS–PAGE gels and transferred onto a polyvinylidene difluoride membrane (Hybond-P; Amersham Biosciences). Sera from K14tgpIV$^{KO}$ or control mice were visualized with HRP-conjugated goat anti-mouse IgG (Bio-Rad Laboratories) and the ECL WesternBright Sirius (advansta).

### Statistical analysis

Statistical significance was assessed by the two-tailed unpaired t test or two-way Anova, and Log-rank test using Prism software (GraphPad).

## Acknowledgements

The authors thank JP Aubry-Lachainaye and C Gameiro for excellent assistance in flow cytometry. We also thank Claire-Anne Siegrist, Floriane Auderset, and Assunta Caruso for valuable discussions and/or technical help. This work was supported by the Swiss National Science Foundation (PP00P3_152951 and 310030_166541 to S Hugues) and the European Research Council (281365 to S Hugues).

### Author Contributions

J Dubrot: conceptualization, formal analysis, investigation, methodology, and writing—original draft.

FV Duraes: conceptualization, formal analysis, investigation, and methodology.
G Harlé: formal analysis, investigation, and methodology.
A Schlaeppi: formal analysis, investigation, and methodology.
D Brighouse: formal analysis and methodology.
N Madelon: formal analysis, investigation, and methodology.
C Goepfert: formal analysis and investigation.
N Stokar-Regenscheit: formal analysis, investigation, and methodology.
H Acha-Orbea: conceptualization.
W Reith: conceptualization.
M Gannagé: conceptualization.
S Hugues: conceptualization, supervision, project administration, and writing—original draft, review, and editing.

## Conflict of Interest Statement

The authors declare that they have no conflict of interest.

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
