## [Reviewer comments · Life Science Alliance]

Life Science Alliance

Absence of MHC-II expression by lymph node stromal cells results in autoimmunity

Juan Dubrot, Fernanda Duraes, Guillaume Harlé, Anjalie Schlaeppli, Dale Brighthouse, Natacha Madelon, Christine Göpfert, Nadine Stokar-Regenscheit, Hans Acha-Orbea, Walter Reith, Monique Gannage, and Stephanie Hugues

DOI: [10.26508/lsa.201800164](https://doi.org/10.26508/lsa.201800164)

Corresponding author(s): Stephanie Hugues, Geneva Medical School

Review Timeline:

Submission Date:	2018-08-17
Editorial Decision:	2018-09-26
Revision Received:	2018-11-23
Editorial Decision:	2018-12-05
Revision Received:	2018-12-06
Accepted:	2018-12-06

Scientific Editor: Andrea Leibfried

Transaction Report:

September 26, 2018

Re: Life Science Alliance manuscript #LSA-2018-00164-T

Prof. Stephanie Hugues
Geneva Medical School
Department of Pathology and Immunology University of Geneva Medical School
1 rue Michel Servet
Geneva 4 CH-1211
Switzerland

Dear Dr. Hugues,

Thank you for submitting your manuscript entitled "Absence of MHC-II expression by lymph node stromal cells results in autoimmunity" to Life Science Alliance. The manuscript was assessed by two expert reviewers, whose comments are appended to this letter. A third report on your work was promised, but we have not received this report and decided to move forward without it. In case we will receive this missing report in the next coming days, I will forward it to you.

As you will see, the two reviewers find your work interesting, but have rather split views. Reviewer #1 raises concerns that could in principle get addressed in a major round of revision, but reviewer #2 thinks that the effects observed could be due to alternative explanations than lack of endogenous MHCII expression by lymph node stromal cells. This reviewer thinks that the work is therefore not publishable, also not a revised version.

Given this input, we have discussed your work in light of the comments of reviewer #2. While we agree that the input provided is formally correct, we decided that publication of a thoroughly revised version of your manuscript is still warranted. We would thus like to invite you to provide a revised version of your manuscript, addressing all concerns raised by reviewer #1 and toning down your conclusions / making room for alternative explanations in the manuscript text to address the concerns of reviewer #2.

Thank you for this interesting contribution to Life Science Alliance. We are looking forward to receiving your revised manuscript.

Sincerely,

-- High-resolution figure, supplementary figure and video files uploaded as individual files: See our detailed guidelines for preparing your production-ready images, <http://life-science-alliance.org/authorguide>

B. MANUSCRIPT ORGANIZATION AND FORMATTING:

Full guidelines are available on our Instructions for Authors page, <http://life-science-alliance.org/authorguide>

*****IMPORTANT:** It is Life Science Alliance policy that if requested, original data images must be made available. Failure to provide original images upon request will result in unavoidable delays in publication. Please ensure that you have access to all original microscopy and blot data images

before submitting your revision.***

Reviewer #1 (Comments to the Authors (Required)):

The authors study the role of MHC-II expression on LN stromal cells (LNSC) on T cell biology. The two first figures characterize and validate a mouse model in which MHC-II expression is selectively abolished in LNSC. The K14Tg-pIVKO clearly allows a selective inactivation of MHC II expression in the LNSC. The next figure show that signs of autoimmunity appear in old (18 mo) animals deficient in MHC-II expression on LNSC. Indeed, CD4 and CD8 infiltration is present in several organs and auto-antibodies are found in old animals. These results indicate that MHC-II expression on LNSC is necessary for self-tolerance. An increase in memory cells and a decrease in Tregs is observed in old animals. A defect in Tregs numbers and activity is observed. Transfer experiments indicate that both an increase in effector and a deficiency in Treg number are found in the Rag-/- hosts expressing normal levels of MHC-II. Tregs proliferation appears to be dependent upon MHC-II expression (fig. 6).

On a whole, the experiments are correctly performed, the validation of the model is correct. However, the number of animals is sometimes quite low and the results are not properly displayed. As stated by the authors themselves, the molecular mechanism(s) involved is/are not deciphered and the study remains very descriptive.

Major criticisms

1. Fig 4D (suppressive activity of Tregs) is not properly performed. These experiments are usually performed by testing several Treg/Tresponder ratios allowing drawing dose response curves. This enables determining whether the number of Tregs required to inhibit the responder is the same between the experimental groups or the plateau of inhibition is modified. As is, the experiment displayed is not demonstrative. How the Tregs would be defective is not explored (Lack of TGF β , IL-10 or Lack of IL2 consumption...)
2. Not any effort is made to determine whether the inflation of Memory T cells is oligoclonal or polyclonal, what are the antigen recognized etc.... What is the mechanistic relationship between the increase in effector cells and the defect in Treg number or function.
3. In the transfer experiments reported in fig. 5, the authors could have used different number of purified effector or Tregs to assess the pathogenic potency of the memory cells harvested from the different genetic background.
4. To allow the reader to better understand the type of distribution observed in each experiments, individual values should be plotted as dots and the histograms should not be used as advocated in a leading journal by Tracey L. Weissgerber , Natasa M. Milic, Stacey J. Winham, Vesna D. Garovic PLoS Biol 13(4): e1002128 DOI: 10.1371/journal.pbio.1002128). This applies to all figures. To take into account the variability between experiments, the authors may want to pool the data from the different experiments and indicate the experiment by using different symbols for each experiment. This will also allow the reader to see that the "ns" observed in fig. 6A probably to a lack of statistical power as only 6 mice were studied (3 mice in two different experiments).
5. The title of Fig. 6 is too strong: instead of showing that enhanced MHC-II expression in LECs promotes Treg proliferation, they show that deletion of MHC-II on LNSC decreases Treg proliferation. Whether the MHC-II increase is directly responsible for the Treg increase is not so clear (cause or consequence). Whether the increased expression of MHC-II in old mice is really caused by IFN-g is not demonstrated (injection of IFN-g can do it, but whether this is the case at steady is not clear in the absence of IFN-GR/IFNG genetic ablation on specific LNSC cells).

Minors

- In fig 6B-D, the authors should indicate in the Y label that the "proliferation (%)" is for the treg. In the legend of Fig. 6, the histograms mention a denominator on CD25+FoxP3+. How is this possible since a staining with an anti-CD25 antibody is never mentioned in the methods section.

Reviewer #2 (Comments to the Authors (Required)):

In the present study, Dubrot et al. aim to investigate the role of lymph node stromal cells (LNSCs) in the maintenance of peripheral T cell homeostasis. Specifically, they address whether the expression of MHC II by LNSCs is of physiological significance. To do so, they use a mouse model (pIVko) where endogenous expression of MHC II by most non-hematopoietic cells, including LNSCs, is abolished as a consequence of lack of CIITA expression in these cells. Because these mice also lack MHC II on cortical epithelial cells of the thymus (cTECs), the authors 'rescue' MHC II on cTECs (and hence positive selection of CD4 T cells) through transgenic expression of CIITA under the K14 promoter (presumed to be cTEC-specific). They found that upon aging, these mice show signs of autoimmunity and immune dysregulation.

The question raised by the authors is definitely of broad interest. Also, the autoimmune manifestations in aging K14tg pIVko mice and alterations in Treg cells are well characterized. However, there is a major concern as to how conclusively these can be ascribed to lack of endogenous MHCII expression by LNSCs as opposed to i) lack of MHCII expression by any other non-hematopoietic cell type in which its expression depends on pIV and/or (ii) altered T cell selection in the thymus. It seems impossible to exclude these caveats, unless one uses an entirely different system that directly eliminates MHCII on LNSCs (or subsets thereof) using conditional MHCII alleles and a specific Cre-driver instead of the contrived pIVkoK14tg model (that abolishes MHCII in a variety of tissues including LNSCs and 'presumably' restores it in cTECs). In that way, a contribution of (i) lack of MHCII on other tissues to disease development cannot be excluded at all, and to exclude a contribution of (ii) aberrant thymic selection would require to conclusively show that the T cell repertoire that is selected in pIVkoK14tg mice is indeed identical to that in WT mice. Previously published data and data presented in this MS concerning 'normal' T cell selection lack the degree of 'resolution' to make this statement (V region usage, CD4 SP number etc.). Unfortunately, investing in a TCR repertoire characterization in pIVkoK14tg mice would be very laborious and still not deal with caveat (i).

For these reasons, it is unfortunately difficult to recommend in favor of publication or to suggest 'reasonable' revisions.

We thank the reviewers for their comments. We have addressed most of the issues, and feel that how manuscript is now greatly improved. Please find bellow a point-by-point answer to the reviewers' comments. All changes have been highlighted in yellow in the revised version of the manuscript.

Of note, we realized a mistake was included in the initial version of the manuscript. The frequency of CD4+ T cells producing IFN γ and IL17 was originally inverted, this has been corrected in the revised version (Figure 3B).

Reviewer #1 (Comments to the Authors (Required)):

The authors study the role of MHC-II expression on LN stromal cells (LNSC) on T cell biology. The two first figures characterize and validate a mouse model in which MHC-II expression is selectively abolished in LNSC. The K14Tg-pIVKO clearly allows a selective inactivation of MHC II expression in the LNSC. The next figure show that signs of autoimmunity appear in old (18 mo) animals deficient in MHC-II expression on LNSC. Indeed, CD4 and CD8 infiltration is present in several organs and auto-antibodies are found in old animals. These results indicate that MHC-II expression on LNSC is necessary for self-tolerance. An increase in memory cells and a decrease in Tregs is observed in old animals. A defect in Tregs numbers and activity is observed. Transfer experiments indicate that both an increase in effector and a deficiency in Treg number are found in the Rag $^{-/-}$ hosts expressing normal levels of MHC-II. Tregs proliferation appears to be dependent upon MHC-II expression (fig. 6).

On a whole, the experiments are correctly performed, the validation of the model is correct. However, the number of animals is sometimes quite low and the results are not properly displayed. As stated by the authors themselves, the molecular mechanism(s) involved is/are not deciphered and the study remains very descriptive.

Major criticisms

1. Fig 4D (suppressive activity of Tregs) is not properly performed. These experiments are usually performed by testing several Treg/Tresponder ratios allowing drawing dose response curves. This enables determining whether the number of Tregs required to inhibit the responder is the same between the experimental groups or the plateau of inhibition is modified. As is, the experiment displayed is not demonstrative. How the Tregs would be defective is not explored (Lack of TGF β , IL-10 or Lack of IL2 consumption...)

We have repeated again the in vitro Treg suppressive assay at different ratios. We observed a difference in the ability of Tregs isolated from control and KO mice only at the ratio 1:5, but not at ratios 1:1 or 1:3, showing that, as suggested by the reviewer, that when sufficient Treg numbers are provided, we don't see impairment in suppressive activity anymore. These new data have been added in Figure 4E. Unfortunately, we could not correlate impaired suppressive Treg functions with differences in their ability to produce IL-10, which was undetectable in the culture supernatant (tested by ELISA not shown).

2. Not any effort is made to determine whether the inflation of Memory T cells is oligoclonal or polyclonal, what are the antigen recognized etc.... What is the mechanistic relationship between the increase in effector cells and the defect in Treg number or function.

LNSC express a broad range of self-antigens. Therefore, our hypothesis is that abrogation of MHCII in these cells will lead to an inflation of polyclonal memory/effector T cells. This is now discussed in the discussion section of the manuscript. We also discuss the fact that several TCR Vb chains were tested by flow cytometry in CD4+ and CD8+ T cells isolated from old controls and knockout mice, and that no difference was observed in the frequency of the different TCR Vb chains expressed (data not shown).

However, we performed additional experiments showing that the frequency of effector CD4+ and CD8+ T cells expressing PD-1, as well as the frequency of CD4+ and CD8+ T cells producing IFN-g, were enhanced in skin LNs from K14tg^{IVKO} compared to K14tg mice. The fact that both CD4+ and CD8+ T cell phenotypes and effector functions were affected in LNs suggests that a suppression by Tregs is lost in LNs from old knockout mice. These data are included in Figure 4B.

3. In the transfer experiments reported in fig. 5, the authors could have used different number of purified effector or Tregs to assess the pathogenic potency of the memory cells harvested from the different genetic background.

We apologize but we could not address this point, since, in order to transfer different numbers of purified T cell populations, it would have require a huge number of 1.5 year old mice that we don't have at the moment, and adoptive transfer experiments in several Rag2^{-/-} mice that require > 4 months.

4. To allow the reader to better understand the type of distribution observed in each experiments, individual values should be plotted as dots and the histograms should not be used as advocated in a leading journal by Tracey L. Weissgerber , Natasa M. Milic, Stacey J. Winham, Vesna D. Garovic PLoS Biol 13(4): e1002128 DOI: 10.1371/journal.pbio.1002128). This applies to all figures. To take into account the variability between experiments, the authors may want to pool the data from the different experiments and indicate the experiment by using different symbols for each experiment. This will also allow the reader to see that the "ns" observed in fig. 6A probably to a lack of statistical power as only 6 mice were studied (3 mice in two different experiments).

Figures were modified accordingly. More experiments were performed for Figure 6A, and we still see that the difference in MHCII expression is significant for the LEC population, with only a trend for BECs, and no difference for FRCs. This is now stated in the result part.

5. The title of Fig. 6 is too strong: instead of showing that enhanced MHC-II

expression in LECs promotes Treg proliferation, they show that deletion of MHC-II on LNSC decreases Treg proliferation. Whether the MHC-II increase is directly responsible for the Treg increase is not so clear (cause or consequence). Whether the increased expression of MHC-II in old mice is really caused by IFN-g is not demonstrated (injection of IFN-g can do it, but whether this is the case at steady is not clear in the absence of IFN-GR/IFNG genetic ablation on specific LNSC cells).

We have change the title of the Figure 6.

We agree with the reviewer that we did not demonstrate that the upregulation of MHCII on LECs in old mice is due to increased IFN-g levels. Indeed, we could not detect the cytokine in LN supernatants. However, the pIV of CIITA, which drives MHCII in LNSCs, is IFNg inducible, suggesting that it could be the case. Alternatively, this could result from an enhanced sensitivity to IFN-g. Accordingly, LECs and BECs express higher levels of IFN-g receptor in LNs from elderly compared to younger mice (New experiments added in Figure 6A).

Minors

- In fig 6B-D, the authors should indicate in the Y label that the "proliferation (%)" is for the treg. In the legend of Fig. 6, the histograms mention a denominator on CD25+FoxP3+. How is this possible since a staining with an anti-CD25 antibody is never mentioned in the methods section.

Tregs were indeed detected based on Foxp3 staining, and no anti-CD25 antibody was used. We apologize for this mistake, and have removed CD25+ from the histogram denominator.

Reviewer #2 (Comments to the Authors (Required)):

In the present study, Dubrot et al. aim to investigate the role of lymph node stromal cells (LNSCs) in the maintenance of peripheral T cell homeostasis. Specifically, they address whether the expression of MHC II by LNSCs is of physiological significance. To do so, they use a mouse model (pIVko) where endogenous expression of MHC II by most non-hematopoietic cells, including LNSCs, is abolished as a consequence of lack of CIITA expression in these cells. Because these mice also lack MHC II on cortical epithelial cells of the thymus (cTECs), the authors 'rescue' MHC II on cTECs (and hence positive selection of CD4 T cells) through transgenic expression of CIITA under the K14 promoter (presumed to be cTEC-specific). They found that upon aging, these mice show signs of autoimmunity and immune dysregulation.

The question raised by the authors is definitely of broad interest. Also, the autoimmune manifestations in aging K14tgpIVko mice and alterations in Treg cells are well characterized. However, there is a major concern as to how conclusively these can be ascribed to lack of endogenous MHCII expression by LNSCs as opposed to i) lack of MHCII expression by any other non-hematopoietic cell type in which its expression depends on pIV and/or (ii) altered T cell selection in the thymus. It seems impossible to exclude these caveats, unless one uses an entirely different system that directly eliminates MHCII on LNSCs (or subsets thereof) using conditional MHCII alleles and a specific Cre-driver instead of the contrived

pIVkoK14tg model (that abolishes MHCII in a variety of tissues including LNSCs and 'presumably' restores it in cTECs). In that way, a contribution of (i) lack of MHCII on other tissues to disease development cannot be excluded at all, and to exclude a contribution of (ii) aberrant thymic selection would require to conclusively show that the T cell repertoire that is selected in pIVkoK14tg mice is indeed identical to that in WT mice. Previously published data and data presented in this MS concerning 'normal' T cell selection lack the degree of 'resolution' to make this statement (V region usage, CD4 SP number etc.). Unfortunately, investing in a TCR repertoire characterization in pIVkoK14tg mice would be very laborious and still not deal with caveat (i).

We agree with the reviewer 2 that our mouse model is not optimal. However, regarding caveat (ii), it is really unlikely that aberrant thymic selection is happening in K14pIV KO mice. Indeed, MHCII in cTECs is not 'presumably' but 'effectively' restored to levels identical to cTECs in WT mice (Figure 1C). In addition, a bias TCR repertoire that would explain the apparition of signs of autoimmunity in knockout mice would more be consequence of impaired negative selection, rather than positive selection, which is even more unlikely given the fact that we don't affect endogenous MHCII expression by mTECs in K14pIV KO mice (mTECs express the pIII of CIITA in addition to pIV and second, the K14 promoter is not active in terminally differentiated mTECs).

Caveat (i) is more of an issue, and we cannot exclude indeed that cells in peripheral tissues contribute to the phenotype we observed in old K14pIV KO mice. However, the fact that we recapitulate our phenotype after transfer of LN T cells into Rag2^{-/-} mice supports our conclusions. In addition, we see an impaired Treg proliferation in LNs few days after upregulation of MHCII in LNSC (after IFNg injection) in both K14pIV KO and Prox1Cre MHCII flox mice, which is in accordance with a role for those cells in impacting self-specific T cells and maintaining peripheral T cell tolerance. We are now discussing this issue, and we propose alternative explanations in the revised version of our manuscript.

December 5, 2018

RE: Life Science Alliance Manuscript #LSA-2018-00164-TR

Prof. Stephanie Hugues
Geneva Medical School
Department of Pathology and Immunology University of Geneva Medical School
1 rue Michel Servet
Geneva 4 CH-1211
Switzerland

Dear Dr. Hugues,

Thank you for submitting your revised manuscript entitled "Absence of MHC-II expression by lymph node stromal cells results in autoimmunity". Reviewer #1 assessed this version again and appreciates the way you addressed the criticisms of the reviewers. We would be thus happy to publish your paper in Life Science Alliance pending final revisions necessary to meet our formatting guidelines:

- please include the statistical tests used in the figure legends
- please provide the manuscript text as a word docx file
- please upload individual figure files
- please make sure that the order of authors is the same in the submission system and on the manuscript file
- please link your ORCID iD to your account, you should have received an email with instructions on how to do so

A. FINAL FILES:

-- High-resolution figure, supplementary figure and video files uploaded as individual files: See our detailed guidelines for preparing your production-ready images, <http://life-science-alliance.org/authorguide>

-- Summary blurb (enter in submission system): A short text summarizing in a single sentence the study (max. 200 characters including spaces). This text is used in conjunction with the titles of

papers, hence should be informative and complementary to the title. It should describe the context and significance of the findings for a general readership; it should be written in the present tense and refer to the work in the third person. Author names should not be mentioned.

B. MANUSCRIPT ORGANIZATION AND FORMATTING:

Full guidelines are available on our Instructions for Authors page, <http://life-science-alliance.org/authorguide>

Sincerely,

Andrea Leibfried, PhD
Executive Editor
Life Science Alliance
Meyershofstr. 1
69117 Heidelberg, Germany
t +49 6221 8891 502
e a.leibfried@life-science-alliance.org
www.life-science-alliance.org

Reviewer #1 (Comments to the Authors (Required)):

The authors have successfully answered the different criticisms. No more comment.

December 6, 2018

RE: Life Science Alliance Manuscript #LSA-2018-00164-TRR

Prof. Stephanie Hugues
Geneva Medical School
Department of Pathology and Immunology University of Geneva Medical School
1 rue Michel Servet
Geneva 4 CH-1211
Switzerland

Dear Dr. Hugues,

Thank you for submitting your Research Article entitled "Absence of MHC-II expression by lymph node stromal cells results in autoimmunity". It is a pleasure to let you know that your manuscript is now accepted for publication in Life Science Alliance. Congratulations on this interesting work.

DISTRIBUTION OF MATERIALS:

Again, congratulations on a very nice paper. I hope you found the review process to be constructive and are pleased with how the manuscript was handled editorially. We look forward to future exciting submissions from your lab.

Sincerely,
